# The Landscape of Anthrax Prevention and Control: Stakeholders’ Perceptive in Odisha, India

**DOI:** 10.3390/ijerph17093094

**Published:** 2020-04-29

**Authors:** Krushna Chandra Sahoo, Sapna Negi, Deepika Barla, Goldi Badaik, Sunita Sahoo, Madhusmita Bal, Arun Kumar Padhi, Sanghamitra Pati, Debdutta Bhattacharya

**Affiliations:** 1Regional Medical Research Centre, Indian Council of Medical Research, Bhubaneswar, Odisha 751023, India; sahookrushna79@gmail.com (K.C.S.); sapnanegi512@gmail.com (S.N.); deepikabarlad@gmail.com (D.B.); dr.goldibadaik156@gmail.com (G.B.); sunitasahoo033@gmail.com (S.S.); balmadhusmita@gmail.com (M.B.); drsanghamitra12@gmail.com (S.P.); 2District Headquarter Hospital, Koraput, Odisha 764020, India; drarunpadhi@gmail.com

**Keywords:** one-health concept, anthrax, zoonotic diseases, veterinary diseases, stakeholders

## Abstract

The prevalence and outbreaks of anthrax are interlinked with the animal-environment-human context, which signifies the need for collaborative, trans-disciplinary and multi-sectoral approaches for the prevention and control of anthrax. In India, there are hardly any shreds of evidence on the role of various stakeholders’ on anthrax prevention and control. Therefore, this study addressed the experiences of various stakeholders on anthrax prevention and control strategies in Odisha, India. A qualitative explorative study was carried out using 42 in-depth-interviews among the stakeholders from health, veterinary and general administrative departments from the block, district, and state level. Two major themes emerged: (1) Epidemiological investigation of anthrax in Odisha, India, and (2) Biological and social prevention strategies for anthrax in Odisha, India. The study emphasizes setting up the surveillance system as per standard guideline, and strengthening the diagnostic facility at a regional medical college laboratory to avoid delay. Moreover, it emphasizes step-up inter-sectoral co-ordination, collaboration and sensitization among health, veterinary, forestry, education, nutrition and tribal welfare departments at all levels in order to reduce the prevalence and control the outbreaks of anthrax in Odisha state. It also recommends raising community literacy, in particular on safe carcass disposal, changing behavior on dead-livestock consumption, and compliance with livestock vaccinations.

## 1. Introduction

Animal, human, and environmental health are interrelated [1,2]. Due to socio-economic fulfillment, the close relation of humans with livestock—conventional agricultural practices and carnivorous food habits promote the emergence and re-emergence of zoonotic diseases, often leading to either endemic or pandemic situations [1,2,3,4,5]. In addition, it was reported that approximately 60% of re-emerging and 75% of emerging infectious diseases in humans came from animal sources [1,2]; which highlights the implications of the one-health concept [4]. Hence, integrated, trans-disciplinary, and multi-sectoral approaches are necessary for zoonotic disease prevention and control [1,2].

The occurrence and outbreaks of anthrax are interlinked with the animal-environment-human context [4,6,7]. It is one of the neglected zoonotic diseases caused by ‘*Bacillus anthracis*’ bacteria, and has life-threatening potential in both animals as well as humans. The transmission of infection occurs through inhalation, ingestion, injection or cutaneous route—spores can remain alive in the soil for decades [4,5,6,7]. Untreated cases lead to high mortality in both livestock and humans [2]; it also has a possible role in bio-terrorism [8].

Anthrax is widespread in many parts of the world [4,5,6,7,8,9]. Globally, there are around 1.83 billion people living within anthrax-risk areas, but most of that population faces little occupational exposure [9]. It is a significant public health threat in developing countries because most of the population is engaged in agronomic and animal rearing practices [10], and the low animal vaccination coverage [2]. Most of the time, individuals engaged in farm or animal husbandry are at greater risk of infection [9]. Both human and livestock vulnerability are spread throughout North America, Africa, and Asia—especially in arid and tropical rural rainfed systems [9,10]. Approximately 63.8 million poor farmers and around 1.1 billion livestock were exposed to anthrax infections [9]. Among south Asian countries it is especially endemic in India and Bangladesh [3]. Notwithstanding this, anthrax monitoring programs including diagnosis and case recording remain inadequate in endemic areas [7,8,9,10].

Several outbreaks have been recorded in India over the past decade, particularly in hilly-terrain regions–mainly in south-eastern India—including Odisha State [3]. In Odisha, about 80% of the population live in rural areas and rely for their livelihood on agriculture or animal husbandry [11]. Moreover, more than 22% of the population of Odisha are scheduled tribes–indigenous groups officially recognized in India as economically poor and socially vulnerable and a majority of them live in hilly and forest areas, and have strong cultural values and practices [11]. Many of the tribal majority districts in Odisha are anthrax endemic [3,12,13,14]. The key reasons behind anthrax episodes in Odisha’s tribal pockets were the conventional butchering practices, unsafe handling and disposal of the contaminated carcasses, the habit of consuming dead animals, and poor vaccination coverage [3,12,14]. The effectiveness of zoonotic disease prevention and control strongly relies on inter-sectoral and inter-departmental collaboration and cooperation at different levels [1,7]. In India, however, there is hardly any evidence on the role of various stakeholders in prevention and control of anthrax. Consequently, this study addressed the experiences of medical, veterinary and general administration officers at blocks, districts, and state levels on anthrax prevention and control strategies in Odisha, India.

## 2. Materials and Methods

### 2.1. Study Settings

The study was conducted in the four endemic districts—Koraput, Malkangiri, Rayagada, Sundargarh—of Odisha, India. In the mentioned districts anthrax is reported in both livestock and humans. Geographical areas and related anthrax burdens in Odisha, India are shown in Figure 1. The study settings are highlighted in deep color in the Odisha map in Figure 1. These districts were purposively selected on the basis of available epidemiological data on anthrax outbreak from the state Integrated Diseases Surveillance System (IDSP).

The study settings represent a rural environment with low population density and low literacy rate [11]. These are among the tribal-dominated districts with more than 50% population being tribal. The Koraput district is located in the eastern region of the state, surrounded by Rayagada in the east, Malkangiri in the south and Chhattisgarh state border in the west. The district has a population density of 157 people per km^2^ and a literacy rate of 49%; the Konda, Parajas and Gadabas are the main tribes. Malkangiri district represents a dense forest area with a population density of 106 per km^2^ and literacy rate of 48%; primarily dominated by the Bondas, Koyas, Porajas and Didayis tribes. In the Rayagada district, having a population density of 137 people per km^2^ and literacy rate 49%, the Kandhas and Soras are the major tribes. Sundergarh is recognized as an industrial district of Odisha with a moderate population density of 216 people per km^2^ and literacy rate of 74%, Munda, Oraron and Kisan tribes being the major tribes [11]. Agriculture is the economic base of these districts and the bulk of the population is engaged in animal husbandry activities.

### 2.2. Study Design and Participants

A qualitative explorative study was carried out. A total of 42 in-depth-interviews were conducted among the stakeholders from health, veterinary, and general administrative departments; they were from the block, district, and state-level officers. The block-level officers were Medical Officer (MO) of community health centers, Block Veterinary Officer (BVO), and Block Development Officer (BDO). The district-level stakeholders were District Medical Officers (DMO), and District Veterinary Officers (DVO). Interviews were also performed with the State epidemiologist, Public Health Director (DPH), State Veterinary Officer (SVO), and an expert from Animal Disease Research Institute (ADRI). The participants’ details characteristics are given in Table 1. Participants were contacted in person or over the telephone before the interview. The interviewees were not offered or paid a payment.

### 2.3. Data Collection

The data were gathered through in-depth interviews (IDIs). The interview guide consisted of the open-ended questions and probes developed based on the Anthrax Prevention Guideline of the Centers for Disease Control and Prevention (CDC) [6]. The interviews were conducted in Odia language. Each interview lasted for 25–40 min for medical officers and veterinary officers, and 15–20 min for administrative officers. The interviews were digitally recorded after obtaining consent. The IDIs were conducted by the third, fourth, and fifth author. We followed the consolidated criteria for reporting qualitative research (COREQ) for reporting. The co-investigators were from diverse educational backgrounds with public health perspective: medical, environmental health, microbiology and nursing.

### 2.4. Data Management and Analysis

The digitally recorded interviews were first transcribed verbatim and then translated into the English language. The data were then subjected to content analysis [14]. Initially, the unit of analysis was selected. Meaning units are the extracts of original transcripts, selected from the interviews and are linked to the aim of the study. The meaning units were firstly condensed to form codes. Related codes were then grouped together into sub-categories and categories. Subsequently, the themes, which demonstrate the core meaning of texts, were identified.

### 2.5. Ethical Considerations

The institutional ethical committee of ICMR-Regional Medical Research Centre, Bhubaneswar, approved the study (ethical approval number: ECR/911/Inst/OR/2017). Before the interview, the purpose of the study was explained to the interviewees and they were informed about their right to withdraw from the study at any point in time. Written consent was obtained from each interviewee. The privacy and confidentiality of the interviewees were maintained throughout the interview.

## 3. Results

Two major themes emerged: (1) Epidemiological investigation of anthrax in Odisha, India (Table 2), and (2) Biological and social prevention strategies for anthrax in Odisha, India (Table 3). The findings presented under each theme with categories, sub-categories and participants’ quotes.

### 3.1. Theme 1: Epidemiological Investigation of Anthrax in Odisha, India

Three categories identified: (1) existing surveillance systems; (2) current laboratory and diagnostic facilities, and (3) outbreak investigation mechanisms.

#### 3.1.1. Category: Existing Surveillance Systems

(a) Sub-category: Burden of Anthrax:

According to the state-level officers, out of thirty districts of Odisha, nineteen districts (highlighted in Figure 1) were reported as endemic for anthrax. In all the nineteen districts, anthrax was diagnosed among livestock, whereas in only four districts—Koraput, Rayagada, Malkangiri and Sundergarh—were human anthrax cases reported. In these three districts, specific blocks, as well as villages, were identified as anthrax hotspots. Koraput is highly endemic for Anthrax, where seven out of fourteen blocks affected frequently reported cases of anthrax. Among the three types of anthrax reported, the cutaneous form was most commonly occurring across the state.

Many participants claimed that anthrax was linked with cultural practices. Consumption of dead-animal contaminated meat was a significant cause for the infection. Tribal communities ‘poor financial circumstances pushed them to eat dead-livestock without any concerns about the consequences. They also indicated the need for intensive surveillance to track and avoid the spread.
“*In Rayagada, Kashipur is a high endemic—specific group of the scheduled caste (‘Dalits’ officially regarded as socially disadvantaged) and tribal communities were consuming dead-cattle meat, which spread infections*”.(DVO)

(b) Sub-category: System and reporting flow:

Many participants had been unaware of the anthrax surveillance system. Several of them outlined the reporting protocol in case of any possible human anthrax contamination. Whenever patients with clinical anthrax signs, usually irregular boils or pustule, visited health centers, they immediately sent the sample to the district laboratory for confirmation; and reports to district IDSP officers.
“*We don’t have active surveillance, if patients have unusual boils, pustules or suspect anthrax, we report it to the district health officers*”.(MO)

The participants acknowledged that an active surveillance system has not been established for reporting. The IDSP under National Health Mission weekly reported the number of anthrax-patients visiting the public health facilities in anthrax endemic districts.
“*Anthrax was not in the surveillance system, we are reporting under IDSP*”.(State Epidemiologist)

Many veterinary officers opined that there is a lack of routine surveillance in livestock; however, the reporting was performed through the National Animal Disease Reporting System (NADRS) portal by BVOs. They were unable to provide detailed information on the process for reporting NADRS flow from village to block/district/state/national level. They perform “sero-surveillance,” in which animal blood smear collected after twenty-one days of vaccination, and referred to ADRI for vaccine efficacy assessment.
“*Hardly any surveillance in livestock. During the outbreak, blood and soil samples from epidemic villages sent to the animal disease diagnostic laboratory; they only assisted during outbreaks, not regularly*”.(DVO)

#### 3.1.2. Category: Current Laboratory and Diagnostic Facilities

(a) Sub-category: Current diagnostics available:

In human infection, blood and swab samples collected and sent to the district laboratory, where they perform the gram staining and refer to the national laboratory for furthermore diagnosis. The district’s laboratories were unequipped for advance diagnosis.
“*The mobile health team investigates the case on the spot, if suspect anthrax; collect the smears from wounds and send to the district laboratory*”.(MO)

The blood and bone samples of suspected dead livestock, and soil samples from the slaughtering-site collected by DVO along with the ADRI team, and refer to the state animal reference laboratory where advanced laboratory facilities available; few samples also referred to the national laboratory.

(b) Sub-category: Challenges in laboratories diagnostics:

The major perceived diagnostic problems were lack of diagnostic facilities at the regional level, the long distance from the state or national laboratory and inadequate transport facilities, which often lead to delays in diagnosis, even during an outbreak. They suggested that regional diagnostic facilities are needed in endemic anthrax districts. They suggest that the regional medical college laboratory in endemic districts may be strengthened. There was a lack of guidance and training among laboratory personnel and sample collectors on safety measures; they stressed periodic diagnostic technique orientation and training including safety measures.
“*We collect the sample from the village—about 70 km from the district; the district sends it to the state—almost 500 km, and the state sends it to the national laboratory; it took at least 20 days to produce the document. Why not the medical college laboratory located in our district strengthens for anthrax, which will minimize the delay in diagnosis*”.(MO)

#### 3.1.3. Category: Outbreak Investigation Mechanisms

(a) Sub-category: History of outbreaks:

The respondents were of the view that the state reported cases of anthrax every year. Over the last decade, they have found about twenty-five outbreaks and registered around 2000 cases. The largest outbreak occurred in Koraput, affecting approximately hundred people and many livestock—mainly with cutaneous infections. In livestock the outbreaks are more common. Tribal populations of the endemic districts were the most vulnerable group. Participants noted the outbreaks occur during pre-monsoon (April–June). The medical officers were observed few human inhalations and gastrointestinal cases. Recently, there were hardly any cases reported by northern districts, but the southern districts are still high risk.

(b) Sub-category: Investigators, investigating mechanisms and action:

The medical officers told that when a patient with painless ulceration or severe anthrax symptoms visits health facilities they suspect outbreaks. An outbreak team including medical officers, veterinary officers, and district laboratory diagnostic staff visits the suspected villages along with community health workers. They investigate the cases, collect the samples and send samples for testing and confirmation. The team was also inquiring community member dietary practices along with clinical review.

The information on anthrax outbreaks is first reported through the health department. Usually, the livestock deaths were under-reported as it happens suddenly and symptoms often go unnoticed. Immediate outbreak investigations and ring-vaccination were conducted in the affected villages and adjacent villages within a radius of five kilometres. The patients were treated with antibiotics, commonly ciprofloxacin or doxycycline, and tetracycline or penicillin was used in livestock. Awareness campaigns for proper disposal of the carcass were conducted. While standard guidelines for the disposal of carcasses were available, the members of the group hardly practice accordingly.
“*After the death of animals, we suggested digging a 6–8 foot pit and burying the carcass with lime. To avoid infection, the site of death and buried soil is disinfected with formalin or acetic acid and the shelter is supposed to sterilized with radiation; it is not followed in practice*”.(BVO)

### 3.2. Theme 2: Biological and Social Prevention Strategies for Anthrax in Odisha, India

Three major categories identified: (1) Provision of vaccine and vaccination, (2) Multi-sectoral stakeholders’ engagement for prevention, and (3) Social security and support.

#### 3.2.1. Category: Provision of Vaccine and Vaccination

(a) Sub-category: Vaccination policies and regulations:

The participants were aware that anthrax vaccination is only available for livestock. There were two ways for vaccination provision: routine vaccination for healthy livestock, and ring vaccination in endemic areas within 5 km of the spot of outbreak, preferred by livestock inspectors during outbreak. The routine vaccination was conducted biannually—during June–July and December–January months. The recommended dose was one-milliliter for large livestock like cow, buffalo and ox, and half-milliliter for small livestock like goats, sheep, and calves. The numbers of vaccinated livestock were reported weekly in NADRS. Government subsidized the anthrax vaccine and administered free-of-cost in endemic regions, as well as a minimal user charge, one to two rupees per livestock in non-endemic areas.
“*No user fees for the vaccine in endemic blocks; one rupee per livestock in other areas*”.(DVO)

(b) Sub-category: Vaccination status including logistics and challenges:

Most of the livestock in the endemic area had been vaccinated, according to the veterinary officers. The cold chain facilities are built for the storage of vaccines in District hospitals and animal dispensaries. The Odisha Biological Product Institute (OBPI)—a satellite unit is located in the state where the anthrax vaccines are made, procured and distributed. The veterinary officers prepare the annual indent and submit the request to OBPI; the OBPI provides timely delivery to the district of the required amount of vaccine, which is subsequently dispatched by District Veterinary Officer to the animal dispensaries. The list was prepared by the livestock inspectors (LIs)—village wise numbers and livestock types need to be vaccinated and sent for approval to the district office. The district office sets the date of vaccination and instructs for vaccination on local LIs. The mobile veterinary units brought the vaccine to the designated village in the icebox, and administered it through home visits. In addition, many participants express that frequent power outages were major barriers to maintaining the cold-chain system, particularly at the block level.

The vaccination time was generally scheduled early in the morning as the cattle went to the nearest forest for grazing about eight o’clock in the morning. Many times the LIs were not able to reach at the vaccination point in a timely way for various reasons. Firstly, the remote location of endemic villages had poor roads and transport connectivity. Second, because of the lack of adequate accommodation facilities, the LIs live far away from endemic villages. Thirdly, the owners of livestock were ignorant of the importance of vaccination and were thus not cooperative. Finally, human resources were inadequate to vaccinate significant quantities of livestock.
“*Many endemic villages are unreachable—poor road connectivity, sometimes need to cross the river. We can’t tell our livestock inspectors were not working properly; in my block, there is a shortage of staff—for 20 Gram-panchayat which constituted about 300 villages; how can only ten number of staff control anthrax!*”.(BVO)

(c) Sub-category: Community literacy and acceptance towards vaccination:

While the vaccines are given free of cost in endemic villages and with minimal cost in other regions, the veterinary officers reported that it was still difficult to fully cover vaccination in livestock. Most of the professionals pointed out the vaccine issues related to community. The frequently perceived cause was a misconception among livestock owners that owing to vaccination cattle that fall-sick or die or decrease productivity. Some of the veterinary officers perceived people getting irritated with vaccinating the livestock for various diseases due to their regular visits to the house. They thought it might be important to incorporate multivalent-vaccine. All most all participants suggested the need for community literacy and engagement for increased vaccine acceptance and anthrax prevention. 

#### 3.2.2. Category: Multi-Sectoral Stakeholders’ Engagement for Prevention and Control

(a) Sub-category: Current scenario on stakeholder’ involvement:

The participants expressed concern about the transmission inter-linkage: animal-environment-human. They acknowledged multi-stakeholders engagement is crucial to prevent Anthrax; however, most of them viewed that except the health and veterinary department at block and district level other department involvements were minimal. At the block level, monthly nodal meetings were conducted where anthrax activities were discussed among the stakeholders especially from health and veterinary. A joint review by the health and veterinary team was carried out during the outbreak, and the document was shared with district administrators.

The participants from all the departments expressed that at the village level there is better inter-departmental coordination through the Gaon Kalyan Samiti (GKS) platform, where a village-level representative from multiple departments are involved. Many participants reported that the community health workers are promoting awareness in GKS meetings; leaflets about anthrax prevention and control distributed among the community members.
“*In Palli Sabha and Gram Sabha (village level convergence meeting), we are promoting the people for preventing anthrax*”.(BDO)

(b) Sub-category: Challenges for stakeholders’ engagement:

Many participants expressed their views on the challenges related to inter-departmental co-ordination. Some of the medical officers felt it was the veterinary department’s duty to prevent Anthrax, as most of the livestock suffer from anthrax. Likewise some of the veterinary officers shared some health practitioners’ non-cooperative attitude. While other departments such as forest, nutrition, education, and tribal welfare have an important role in the prevention of anthrax, their contribution has been negligible, according to the respondents’ opinion.
“*Definitely, the health and veterinary departments play crucial role for anthrax prevention; likewise, there is a need for active participation of the other departments like Integrated Child Development Scheme (ICDS), education, forestry, Integrated Tribal Development Agency (ITDA) and Non-Governmental Organizations (NGOs) to control the spread of anthrax*”.(MO)

Some health professionals believed that the veterinary department had not provided community members with appropriate instructions for carcass disposal. They also perceived a lack of proper knowledge among community members about Anthrax. Few of the professionals highlighted the need for participation and orientation among local mass media and non-governmental organizations, especially in endemic blocks, as they have a better understanding of regional background and language, and are familiar with local culture.
“*The local media, PRI (Panchayat Raj Institution) members and traditional/spiritual healers also have an important role to change community behaviors*”.(BDO)

One professional viewed the need to develop inter-state coordination due to the migration of livestock. The respondents suggested that periodic information on vaccination among livestock owners is important through interdepartmental co-ordination.
“*Full coverage of vaccination among animals is impossible without the involvement of block, medical and district administration*”.(DVO)

#### 3.2.3. Category: Social Security and Support

(a) Sub-category: Insurance provisions:

The veterinary officers acknowledged the centrally sponsored insurance policy for vaccinated livestock under the National Livestock Mission (NLM). The respondents revealed that the scheme provides a protection mechanism for the livestock owners against any uneventful death—due to disease, natural disasters or lightning. They were providing the insured amount through the insurance company. The insured amount of livestock was decided on the basis of the market price. The livestock owners pay the fixed premium amount on annual installment. In uneventful death, the insurance company pay the insured amount to the owner however there were hardly any insurance provisions for non-vaccinated livestock.

(b) Sub-category: Compensation facilities:

In case of anthrax-related death there were insurance schemes for livestock owners. A compensation of two thousand rupees was given from district administration fund to the livestock owners for carcass disposal, but some participants disagreed with such provision. Most participants considered the deep-rooted cultural traditions to be the explanation behind dead-livestock eating. They also proposed improving tribal food security schemes to reduce dead-livestock intake. They suggested including compensation facilities in endemic-region in anthrax prevention strategies.
“*It’s hard to bring about improvement in the cultural tradition of consuming dried and preserved meat from dead animals; their habit of eating dead cattle meat for centuries. To change their attitude we should improve the tribal food-security schemes*”.(State Epidemiologist)

## 4. Discussion

To our knowledge this is India’s first study to address the perspectives of the various stakeholders on anthrax prevention and control. The views indicated that, in order to prevent anthrax, it is important to develop an active surveillance system for reporting cases in humans and livestock, to strengthen regional medical college laboratories in order to avoid delays in diagnosis, and to train laboratory personnel on safety measures. In addition, it is critical that all related departments must be engaged in anthrax prevention and control strategies, and that community literacy on Anthrax be improved. In Table 4, details on strengths, limitations, perceived suggestions and policy implications are presented.

Odisha state reported many outbreaks of both livestock and human anthrax infection in the last few years. Nine cutaneous anthrax outbreaks occurred in the district of Koraput with a case fatality rate of two percent from 2010–2014; primarily the prevalence among males was higher than females, most of them belonging to scheduled tribes; they rely on forest and animal products for their subsistence—often contact with dead animals and were consuming dead livestock meat [3,4,5,6]. Community participation plays a vital role in achieving the success and long-term effect of any health intervention. Therefore, along with health education the tribal food security scheme must be strengthened. This clause appears to allow people to report animal deaths and to stop consuming meat from dead animals. In addition, inadequate public health infrastructure in endemic districts exposes to higher anthrax risk [3,13,14,15,16].

A study in Zambia confirmed that anthrax is mainly related to cultural factors [5]. Many studies have identified living near forests and slaughtering and handling the infected carcasses as the most common causes of human anthrax infections and outbreaks [17,18,19]. Evidence across the globe reveals a lack of veterinary services, and inadequate healthcare workforce and an acute shortage of vaccines and delayed reporting of anthrax outbreaks in animals as major challenges related to outbreaks [3,20,21,22]. In order to strengthen the outbreak response, it was suggested that anthrax must be considered as a differential diagnosis in the case of painless ulceration [4,22]. In addition, the medical curriculum should incorporate short-term training on zoonotic and one-health concept including anthrax [23]. However, the main components for the prevention of any zoonotic diseases are robust laboratory facilities and effective surveillance systems [24].

Proper diagnosis is one of the major components for treatment, prevention, and control of anthrax. Confirmatory laboratory situated far away from the endemic districts. In addition, limited transport facilities are available in endemic districts; it has been difficult to transport the sample timely to the national laboratory, which is similar with the observations from other studies in Asia and Africa [3,15,16,25]. A study in China revealed that timely diagnosis is important to inform epidemic prevention departments for an outbreak investigation in order to control the epidemic [7]. Likewise, an earlier study in Ethiopia highlighted the importance of early diagnosis and care for the burden of morbidity and mortality—the undiagnosed cases spread rapidly in the environment [15]. Therefore, in resource-poor settings, the health and veterinary professionals working in remote areas or anthrax endemic regions must be trained for clinical identification of the cases [15]. A comprehensive database including information on livestock and human diagnostic systems as well as environmental monitoring components are an integral part of the surveillance system [24].

Factors such as disease burden, surveillance status, social security and funding were the priorities of the initial implementation agenda [2]. An effective surveillance program will lay the groundwork for effective anthrax control and prevention. This can be improved by creating an organized monitoring framework for all relevant service providers, program managers and key decision-makers in collaboration [2,16,26]. The previous studies revealed that it was also important to generate evidence from current diagnostic facilities and opinions of stakeholders from outbreak investigations training sessions to strengthen the surveillance system [2,16,26]. A study in Ethiopia showed that joint surveillance of zoonotic diseases has a benefit, and suggests the establishment of a specific surveillance framework and guidelines for implementation [16]. A previous study in Georgia using the national surveillance data of anthrax recommends that supervision on carcasses disposal and disinfection of soils, and participatory health education are the major strategies to prevent anthrax [26]. Moreover, the deaths due to anthrax have an economic impact on the livelihood of livestock owners [3]. Therefore, security and support are other important aspects of enhancing anthrax surveillance. Besides this, emphasis must be laid on inter-sectoral collaboration and community education regarding proper carcass disposal, use of personal protective measures, livestock vaccination and timely reporting of animal deaths [17,27].

It was found that routine vaccination policy including ring-vaccination is one of the better strategies for prevention and control of anthrax [28,29,30]. However, the participants of our study claimed that in-spite zero-cost vaccination policy in endemic regions, still the coverage is very low. The low-uptake of livestock vaccination was strongly associated with social and cultural factors rather than economic [30,31,32]. The reluctance and mistrust among livestock owners was the key barrier for vaccination [5]. Conversely, a study in Bangladesh showed that lack of logistics and veterinary staffs were the other reasons behind poor vaccination among livestock [19]. A study conducted in Bolivia suggested the implication of innovation diffusion theory using the concept of one-health to increase the vaccination coverage [30].

One Health is a collaborative, trans-disciplinary, and multi-sectoral approach that focuses on local, state, national, and global collaboration to understand animal-environment-human interrelationships [2,8]. Multi-sectoral cooperation plays a crucial role in the response to outbreaks, as well as in the control and prevention of anthrax; which helps to improve disease-related activities by recognizing and resolving the lacunae in current activities [32]. The involvement of all relevant sectors should coordinate and collaborate especially in endemic districts in order to reduce the disease burden [18,28,29,30,31,32,33,34]. A previous research stressed the value of public-private collaborations for the prevention of anthrax [35,36], which our study participants also viewed.

To enhance the trustworthiness of the study, the triangulation of data sources and analysis were followed. The information was collected from multiple stakeholders like health, veterinary and general administration at various levels such as blocks, districts, and state. During data analysis, to avoid misinterpretation of the findings both English and Odia transcripts were used; in complex cases, the digital recorded versions were looked up. In this study, the authors were from a different professional: research, clinical and academic, and educational: medical, nursing, microbiology, environment and social science backgrounds with experience in public health, which increase the conformability. Furthermore, member check interpretation and discussion of the findings with four study participants was conceded. Although this study was conducted in Odisha, the implications of the findings will be applicable to other states of India, as well as helpful for similar settings in other parts of the world. Together with health, veterinary and general administrative stakeholders, other stakeholders also play a key role in prevention and control of Anthrax. Nevertheless, the limitation of this study was that it did not include peripheral groups—the departments of education, nutrition, and tribal welfare.

## 5. Conclusions

The study highlights the need to set-up the surveillance system as per standard guideline, and to reinforce/restore the diagnostic facility at a regional medical college laboratory to avoid delay. Moreover, it emphasizes step-up inter-sectoral co-ordination, collaboration and sensitization among health, veterinary, forestry, education, nutrition, and tribal welfare departments at all levels in order to reduce the prevalence and control the outbreaks of anthrax in Odisha state. It also recommends increasing community literacy especially on safe carcass-disposal, changing behavior on consumptions of dead-livestock, and compliance towards livestock vaccinations. This study suggests the need for the quantitative studies to determine the magnitude of the issues to prevent the anthrax outbreak in endemic regions.

## Figures and Tables

**Figure 1 ijerph-17-03094-f001:**
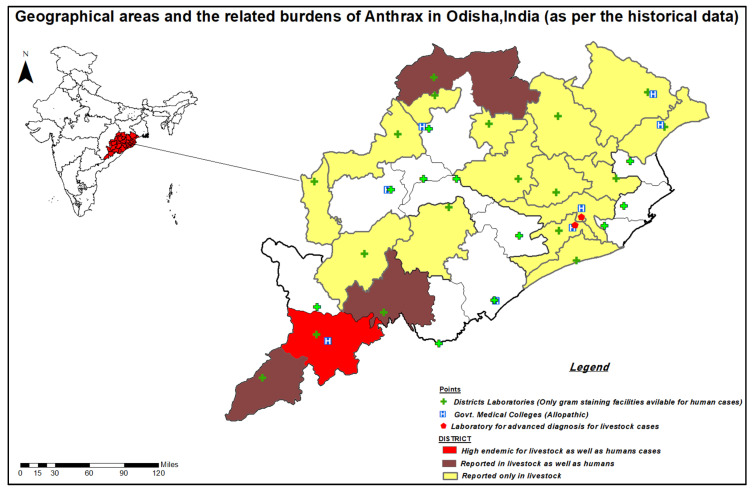
Geographical areas and the related burdens of Anthrax in Odisha, India.

**Table 1 ijerph-17-03094-t001:** Characteristics of the study participants.

Level	Health Department (*n* = 18)	Veterinary Department (*n* = 18)	General Administrators (*n* = 6)
Block (*n* = 12)	Medical Officer (*n* = 12)	Block Veterinary Officer (*n* = 12)	Block Development Officers (*n* = 6)
District (*n* = 4)	District Medical Officers (*n* = 4)	District Veterinary Officers (*n* = 4)	-
State (Odisha)	Epidemiologist (*n* = 1) Director Public Health (*n* = 1)	State Veterinary Officer (*n* = 1) Expert, Animal Disease Research Institute (*n* = 1)	-

**Table 2 ijerph-17-03094-t002:** Theme—Epidemiological investigation of anthrax in Odisha, India.

Theme 1:	Epidemiological Investigation of Anthrax in Odisha, India					
Categories	Existing Surveillance Systems		Current Laboratory and Diagnostic Facilities		Outbreak Investigation Mechanisms	
Sub-Categories	Burden of Anthrax	System and Reporting Flow	Current Diagnostic Available	Challenges in Laboratory Diagnosis	History of Outbreaks	Investigators, Investigating Mechanism and Action
Codes	In animals; In humans; Cutaneous form; Endemic village; Endemic block.	Unusual boils; Passive surveillance; IDSP; Weekly reporting; NADRS portal; Sero-surveillance.	Human-blood and wound swab; Animals- soil, bone and blood sample; DHH—not equipped; ADRI—advanced facilities.	Poor diagnostic facilities; Delayed results; Poor transport; Periodic orientation; Training.	20–25 outbreaks; 2000 suspects; Tribal communities; Seasonal occurrence; Kumbhikari block; Cutaneous form.	Eschar sign; Outbreak team; Home visits; Ring vaccination; Antibiotic prophylaxis; Carcass disposal guidelines; Awareness campaign.

Note: ADRI—Animal Disease Research Institute; DHH—District Headquarter Hospital; IDSP—Integrated Diseases Surveillance System; NADRS—National Animal Disease Reporting System.

**Table 3 ijerph-17-03094-t003:** Theme 2—Biological and social prevention strategies for anthrax in Odisha, India.

Theme 2:	Biological and Social Prevention Strategies for Anthrax in Odisha, India						
Categories	Provision of Vaccine and Vaccination			Multi-Sectoral Stakeholders’ Engagement for Prevention		Social Security and Support	
Sub-categories	Vaccination Policies and Regulations	Vaccination Status Including Logistics and Challenges	Community Literacy and Acceptance towards Vaccination	Current Scenario	Challenges for Stakeholders’ Engagement	Insurance Provision	Compensation Facilities
Codes	Routine vaccination; Ring vaccination; Biannually; Service charge; Sub-cutaneous route.	Coverage; Vaccine storage; Purchase mechanism; Transportation—state to the district, block to village; Vaccine point; Other logistic requirements; Quality maintenance.	Misconception; Non-cooperative behavior; Annoyance; Multivalent vaccine.	Monthly nodal meetings; Weekly Gaon Kalyan Samiti (GKS) meetings; Vaccination campaigns; Veterinary officers training.	Poor veterinary access IEC activities Manpower Interstate coordination Outbreak warning Joint outbreak guidelines	Centrally sponsored National Livestock mission Vaccinated animals Fixed premium amount Uneventful death	Carcass disposal District fund Food security scheme Endemic blocks Attitude and behavior change

**Table 4 ijerph-17-03094-t004:** Summary of findings—Strengths, limitations, perceived suggestions, and policy implications.

Domain	Strengths	Limitations	Perceived Suggestions	Policy Implications
Surveillance	Included in the Integrated Disease Surveillance System (IDSP) and weekly reporting the human cases; Sero-surveillance in livestock.	No active surveillance exist in humans as well as in livestock/animals; Under-reporting of deaths and its’ reasons among livestock.	Need for establishing an active surveillance system for reporting cases in human and livestock.	Development of surveillance system as per the CDC framework.
Diagnosis	Advanced diagnostic facilities at the state level; Basic diagnostic facilities—gram staining at districts level.	Lack of advanced diagnostic facilities at the regional level and inadequate transportation from districts to state; results in delayed diagnosis and reporting; Lack of periodic orientation on safety measures.	Strengthen regional medical college laboratories for anthrax diagnosis; Periodic training on safety measures among laboratory staff.	Capacity building of regional medical college laboratories for diagnosis.
Outbreak	A joint investigation by health and veterinary departments; Ring-vaccination among livestock; Awareness campaign during the outbreak.	Less involvement in education, nutrition, tribal welfare, and forestry departments.	Demand for the involvement of all relevant departments.	Literate the community on standard carcass disposal practice.
Vaccination	Vaccines are locally produced and timely supplied; Weekly reporting of vaccination to NADRS; Free of cost vaccination in endemic blocks; Availability of cold chain facilities at district hospitals and dispensaries; Animal vaccination through home visits	Frequent power outage at remote dispensaries; Misconception and non-cooperative attitude of livestock owners on vaccination; Inadequate manpower; Difficult to reach remote areas.	Introduction of multivalent vaccines for animals; The need for community literacy for vaccination acceptance; Institutional accommodation facilities for veterinary staffs.	Community literacy on vaccination; Solar-power support at vaccine storage point in remote areas.
Inter-departmental coordination	Active involvement of veterinary and health departments; Monthly nodal meetings at the block level; GKS meeting at the village level.	Negligible contribution of other departments like forestry, education, and ITDA; Lack of proper awareness activities on anthrax among community members.	Need for the involvement of mass- media, and NGOs; Need to develop interstate- coordination due to the migration of infected livestock.	Strengthen multi-stakeholder participation; Enhance community literacy through all possible departments.

Note: CDC—Centers for Disease Control and Prevention; GKS—Gaon Kalyan Samiti; ITDA—Integrated Tribal Development Agency; NADRS—National Animal Disease Reporting System; NGOs—Non-Governmental Organizations.

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
