# Peer review of "The Landscape of Anthrax Prevention and Control: Stakeholders’ Perceptive in Odisha, India"

_ijerph, 2020, doi:10.3390/ijerph17093094_

Round 1
Reviewer 1 Report
This work is very interesting but could be so much more so with the addition of certain baseline information and substantiation of the assertions made by the authors. Addition of maps showing the geographic areas discussed along with pertinent prevalence and incidence historical data for the regions and locations of laboratory resources would add the baseline mentioned above and generously provide the relevance to the qualitative study conducted. It is not enough to gather opinions and perspectives to then make assertions of need, particularly when asking for scarce resources to implement enhanced surveillance programs and diagnostic services. Also, the authors have neglected to look harder, deeper, and more broadly as similar studies and papers have been conducted in developing areas of the world.
With a bit more work, background, and sound insight, this could be a very useful and informative paper. As it stands now, it is a loose collection of opinions and assertions aimed at garnering resources.
Reviewer 2 Report
MS 742478 presents the results of a study of stakeholder knowledge, attitudes and behaviors (KABs) toward anthrax prevention and control in Odisha, India. This was an exploratory study, and strictly qualitative in nature. In-depth interviews of 42 stakeholders from health, veterinary, and general administrative departments at differing government levels provided the data. I offer the following comments and suggestions on the manuscript.
Design: The study design seems appropriate to meet study objectives.
Data collection: There are no actual numeric data (tables, etc.) presented. This is probably the greatest shortcoming of the paper. Descriptive, quotes from some of the respondents are offered to indicate typical responses to questions. The study did not include peripheral groups—education, nutrition, tribal welfare—or livestock owners.
Data analysis: There was no statistical or other analysis presented. The study identified a number of shortcomings at all levels in the survey. Diagnostics are either not available or there are long delays in getting results. There is a lack of coordination between agencies and levels. There is a lack of literacy/understanding of anthrax safety issues.
Conclusions: The authors identify a series of measures that can be taken to improve anthrax prevention and control in the region. This is a significant first step in the right direction. However, documenting the impact of any changes in the system will require quantitative measures well beyond the scope of this study. It is not clear if the resources exist for such an effort. To do a quantitative study of this issue would likely require considering a massive number of variables: Current & historical anthrax activity, caste/tribe, education level, geography (accessibility, landscape), and local/regional history.
I have made a few additional notes and comments on the manuscript, and those can be seen by clicking on the Comments option on the right side of the Adobe reader.

Round 2
Reviewer 1 Report
The rewrite of this manuscript is extensive and improves the quality tremendously. Well done.